# An Analysis of the Differences in Vulnerability to Climate Change: A Review of Rural and Urban Areas in South Africa

**Leocadia Zhou** [1]**, Dumisani Shoko Kori** [1,*] **, Melusi Sibanda** [2] **and Kenneth Nhundu** [3]

1  Risk and Vulnerability Science Centre, Faculty of Science and Agriculture, Fort Hare University, P. Bag X1314, Alice 5700, South Africa
2  Department of Agriculture, University of Zululand, P. Bag X1001, KwaDlangezwa, Empangeni 3886, South Africa
3  Economic Analysis Unit, Agricultural Research Council, P.O. Box 8783, Pretoria 0001, South Africa
*  Correspondence: dkori@ufh.ac.za or d_shoko@yahoo.com

**Abstract:** Evidence is unequivocal that rural and urban areas in South Africa are vulnerable to the impacts of climate change; however, impacts are felt disproportionately. This difference in vulnerability between rural and urban areas is presently unclear to guide context-based climate policies and frameworks to enhance adaptation processes. A clear understanding of the differences in vulnerability to climate change between rural and urban areas is pertinent. This systematic review aimed to explore how vulnerability to climate change varies between rural and urban areas and what explains these variations. The approach was guided by the Intergovernmental Panel on Climate Change vulnerability framework incorporating exposure, sensitivity, and adaptive capacity dimensions integrated into the Sustainable Livelihood Framework. The review used 30 articles based on the search criteria developed. The findings show differences in vulnerability to climate change between rural and urban areas owing to several factors that distinguish rural from urban areas, such as differences in climate change drivers, infrastructure orientation, typical livelihood, and income-generating activities. We conclude that vulnerability varies with location and requires place-based analyses. Instead of blanket policy recommendations, localized interventions that enhance adaptation in specific rural and urban areas should be promoted.

**Keywords:** exposure; sensitivity; adaptive capacity; climate change; rural and urban areas; differences in vulnerability; South Africa

## 1. Introduction

South Africa's rural and urban areas face several challenges related to climate change. The challenges highlight a vulnerability to climate change in rural and urban settings. Scholars and organisations dealing with climate change use the term "vulnerability to climate change" [1]. The approaches used to conceptualise the term fall into three categories [2]. The first and second approaches are the "starting point" [3,4] and the "end-point", respectively [3,4]. The "starting point" approach considers that vulnerability is generated by multiple factors and processes while the "end-point" approach considers vulnerability as the residual of climate change impacts [5]. The third approach is the Intergovernmental Panel on Climate Change (IPCC) approach [6], which supersedes the first two approaches in that it is an integrated approach that considers the external biophysical dimension (exposure) and the social dimension (sensitivity and adaptive capacity) [7]. Thus, the extent to which a system will adapt to climate change is influenced by the level of exposure and sensitivity to climate change impacts [8]. Exposure is how a system or community experiences climate-induced, environmental, socio-political, or external, stress [9]. Sensitivity refers to the level of resilience an individual, a system, or a community possesses to withstand climate changes [10]. Adaptive capacity refers to the ability of a system or a community

to respond proactively and positively to stressors or opportunities brought by climate changes [11].

Thywissen [12] established that vulnerability changes over time and across geographical spaces, e.g., rural and urban contexts. This fact may represent that vulnerability to climate change is location specific and spatially variable [13], with some areas more vulnerable than others. South Africa's rural and urban areas are highly vulnerable to the impacts of climate change [14], although the magnitude of vulnerabilities may vary across these different contexts. Therefore, supportive policies and frameworks that are context-specific are needed to enhance climate adaptation processes [8] in various environments. This recommendation aligns with the United Nations (UN) Sustainable Development Goal (SDG) Number 11, which targets establishing sustainable cities and communities and aims to support positive economic, social, and environmental linkages between rural, urban, and peri-urban areas. These linkages are essential for governments to respond effectively to climate change which exacerbates the vulnerability of communities.

While definitions of rural and urban in South Africa are complex and varying [15] with no standardized definition [16,17], some elements can be adapted and applied across different contexts, as observed by World Vision [16]. These elements include settlement types [17], population density, administrative bodies and infrastructure concentration, and common livelihood and income generation activities [16]. In South Africa, using population density, the Limpopo Province is considered the most rural with the highest percentage of the rural population (90%). In contrast, Gauteng Province is the least rural [18].

This paper adopted some of the above elements to contextualise the "rural and urban" definitions of South Africa. Thus, an urban area refers to both metropolitan and non-metropolitan formal areas and informal urban areas characterised by high population density, prominence of administrative structures, and diverse livelihood and income generation activities. Urban informal areas also include areas close to cities and towns known as peri-urban areas. On the other hand, rural areas have relatively low population densities, low to no presence of administrative structures, and livelihood and income generation options are predominantly centered on agriculture. The definitions explained above show that rural and urban areas are different in several factors, which could be drivers of differences in vulnerabilities to climate change.

In the context of this paper, this suggests that there may be differences in terms of vulnerability to climate change between rural and urban areas. Nonetheless, little is known about the level of vulnerability and associated differences between rural and urban areas in South Africa. Several studies have assessed the vulnerability to climate change in South Africa; however, the focus is either on rural or urban contexts separately. For example, Long and Ziervogel [19] tracked the progress of vulnerability assessments in South Africa's urban areas. Samuels et al. [20] assessed the climate vulnerability of an indigenous community in the communal areas of the arid zones of South Africa. Studies assessing the vulnerability of rural and urban areas while unpacking existing variations, especially in South Africa, are scarce. The few studies that attempt to do so, for example, the study from Abrams et al. [21], do not explicitly state the differences in vulnerability between rural and urban areas. This lack of understanding about how vulnerability to climate change varies from rural to urban environments leads to a deficiency in supportive policies and frameworks needed to enhance climate adaptation processes [8]. This situation compromises the United Nations (UN) Sustainable Development Goal (SDG) Number 11 which aims to establish sustainable cities and communities through positive economic, social, and environmental linkages between rural, urban, and peri-urban areas. These linkages are important for governments to respond effectively to issues like climate change which exacerbates the vulnerability of communities.

Therefore, this review sought to explore if there are differences in vulnerability to climate change between rural and urban areas in South Africa. The intention was to pull together vulnerability assessments made in rural and urban areas or both environments into a combined integrated analysis while establishing differences in vulnerability. It was

envisioned that the review would give a deeper understanding of the unique vulnerability differences to climate change between rural and urban areas towards enhancing resilience to climate change in South Africa. The review's findings are intended to inform policymakers to allocate resources accordingly and establish technical aspects of planning human settlements efficiently in response to the specific needs of different rural and urban areas. Such a move would ensure that people residing in more vulnerable areas are better informed to cope with the challenges posed by climate change, thus enhancing their resilience.

## 2. Materials and Methods

This review aimed to explore differences in vulnerability to climate change in South Africa, focusing on rural and urban settings. This was achieved through a rural–urban dichotomy using a three-step review process. We identified the location of vulnerability studies conducted in South Africa, the primary climate-related shocks/hazards that enhance exposure, and the sensitivity and adaptive capacity of rural and urban areas. We also discussed the differences between such vulnerabilities. The procedure followed to achieve this is detailed in the subsequent section.

### 2.1. Search Strategy, Search Terms and Data Sources

The review adopted a systematic review of the literature on vulnerability to climate change, focusing on rural and urban areas in South Africa. Literature sources were extracted from Google Scholar, JStor, and Science Direct databases. Literature sources included full research articles, review papers, and theses addressing vulnerability to climate change in South Africa. Grey literature or unpublished material was excluded. Literature sources were limited to those published in the English language.

An initial search in Google Scholar, JStor, and Science Direct that used the search terms "vulnerability to climate change in South Africa", "vulnerability to climate change in rural areas in South Africa", and "vulnerability to climate change of urban areas in South Africa" yielded a total of 1802 articles. These articles were further screened using a custom range time frame from 2010 to 2021. Only 253 articles were retained. Title and abstract screening were conducted where articles with search terms "vulnerability to climate change in South Africa" in the title and abstract were selected. Only 34 articles were retained. As a final step, an assessment of full-text articles was conducted. Articles were screened based on whether they addressed the key theme of the review, the vulnerability of either rural or urban areas to climate change in South Africa. Only 30 articles were retained for the final review. Figure 1 summarises the article screening process.

Table 1 presents the breakdown of literature sources included in the final review according to type, category, and province. Literature sources included in the review were full primary research articles, review papers, and theses related to the vulnerability of rural and urban areas to climate change in South Africa. Some literature sources assessed vulnerability to climate change for either rural or urban areas, while others assessed vulnerability in both settings. Studies reviewed included five of South Africa's nine provinces, namely Limpopo, Mpumalanga, KwaZulu-Natal, Eastern Cape, and Western Cape. These provinces were in line with the definition of the rural and urban setting in South Africa. Hence, they were appropriate for the study. In some cases, one study assessed vulnerability in two or more provinces. A few assessed the vulnerability to climate change at the national level.

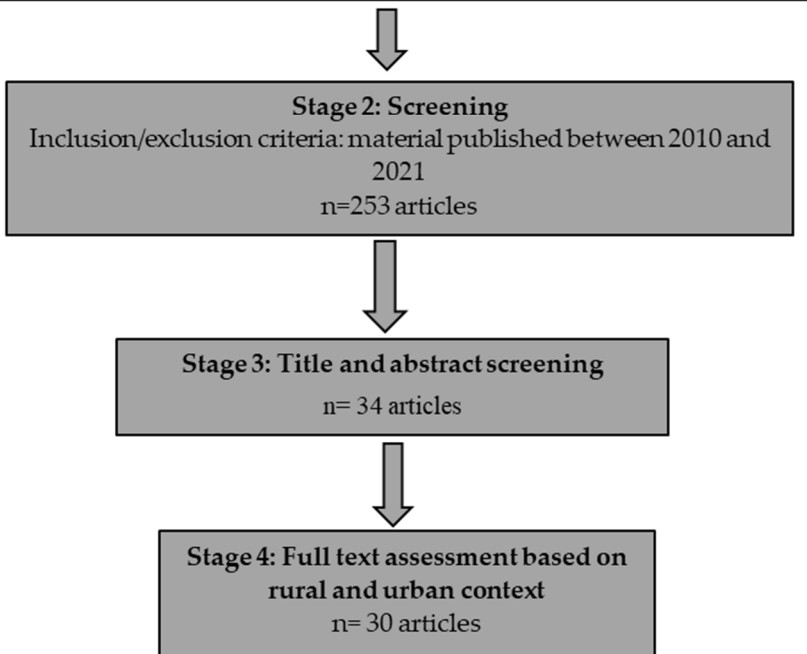

**Figure 1.** Search procedure and screening process. Source: Authors' conceptualisation.

**Table 1.** Breakdown of literature sources included in the review.

| | | |
|---|---|---|
| | Full research articles | 22 |
| Type of literature source | Review papers | 4 |
| | Theses/dissertations | 4 |
| | Rural | 16 |
| Category | Urban | 8 |
| | Both | 6 |
| | Limpopo, | 9 |
| | Mpumalanga, | 1 |
| | KwaZulu-Natal | 10 |
| Province | Eastern Cape | 3 |
| | Western Cape | 3 |
| | Countrywide | 7 |

Source: Authors' analysis derived from the systematic search.

Figure 2 illustrates the spatial coverage of climate change vulnerability research in South Africa. Most of the climate change vulnerability studies reviewed were conducted in Limpopo and KwaZulu Natal Provinces, as represented in dark blue.

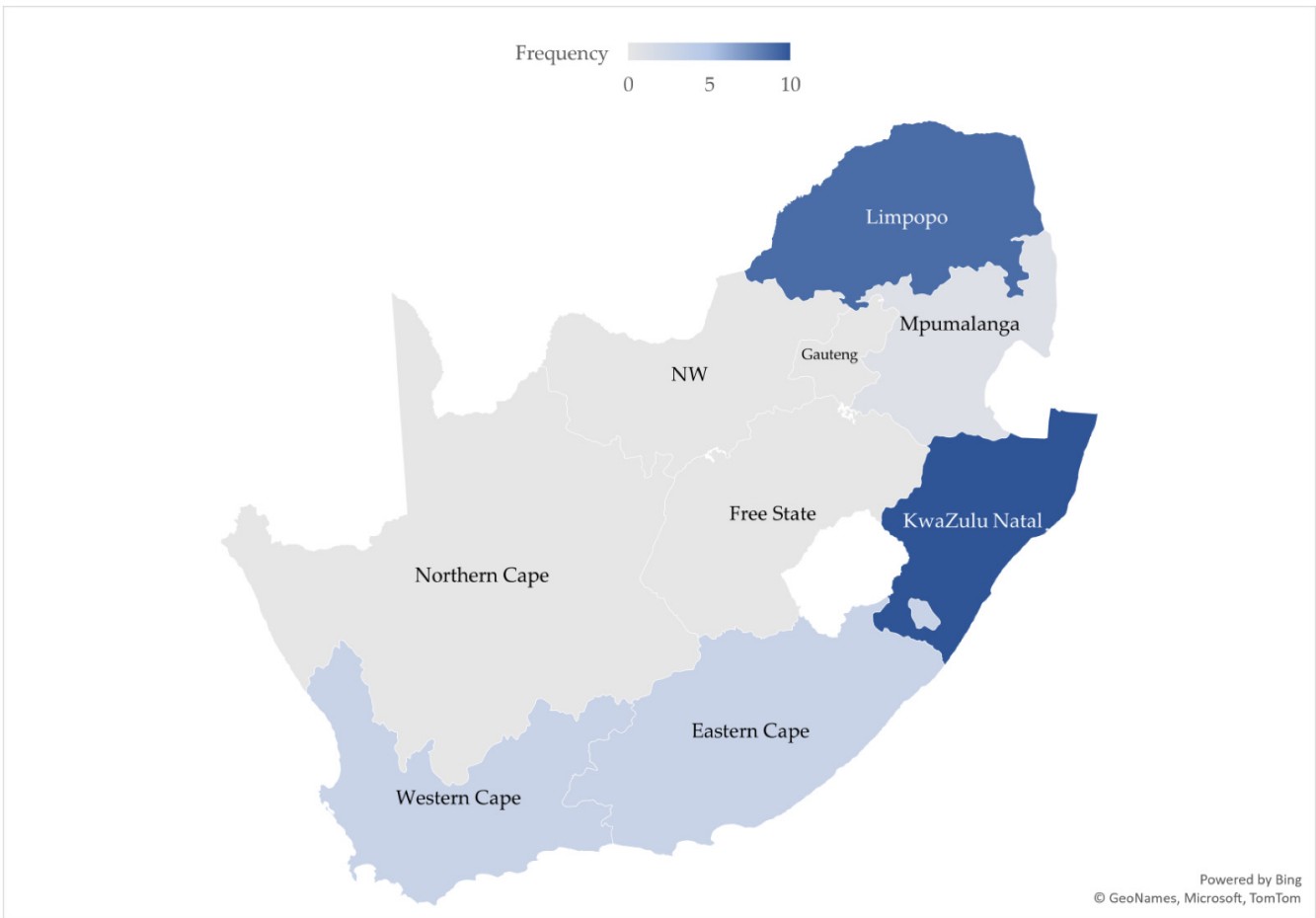

**Figure 2.** Spatial coverage of climate change vulnerability research in South Africa. Source: Authors' analysis derived from the literature sources included in the review.

### 2.2. Conceptual Framework

The dependent variable for this review was the differences in vulnerability to climate change between rural and urban areas. The term "vulnerability" can be conceptualised differently and in numerous ways. For this review paper, we adopted the IPCC conceptualisation of the definition of vulnerability to climate change stating that it is "the degree to which a system is susceptible to, or unable to cope with, the adverse effects of climate change, including climate variability and extremes" [22]. As such, the approaches used in this review were also within this conceptualisation.

Understanding vulnerability within the rural and urban contexts as used in this paper is informed by the perception that vulnerability is an outcome of the interaction between social vulnerabilities and the environmental risk and stresses arising from climate change, as explained in Report No. 4 of the Long-Term Adaptation Scenarios (LTAS) flagship research programme. In addition, the review adopted insights from the Hazards of Place model [23]. The idea of "place" was incorporated into the review to provide a spatial perspective in understanding the dynamics of vulnerability between rural and urban areas in South Africa.

In this context, vulnerability is viewed as the interaction of a system's underlying exposure and sensitivity to climate variations and changes and its ability to adapt. Therefore, vulnerability to climate change for a system or community is determined by exposure, sensitivity, and adaptive capacity [24], as illustrated in Figure 3. Vulnerability is a function

of the nature, magnitude, and rate of climate variation and change to which a system is exposed and its sensitivity and adaptive capacity. This function is illustrated as follows:

$$V = f(E, S, AC)$$

where V = Vulnerability of an urban or rural system or community to climate change variability

E = Exposure of a system or community (urban or rural) to climate change

S = Sensitivity of a system or community (urban or rural) to climate change

AC = Adaptive capacity of a system or community (urban or rural)

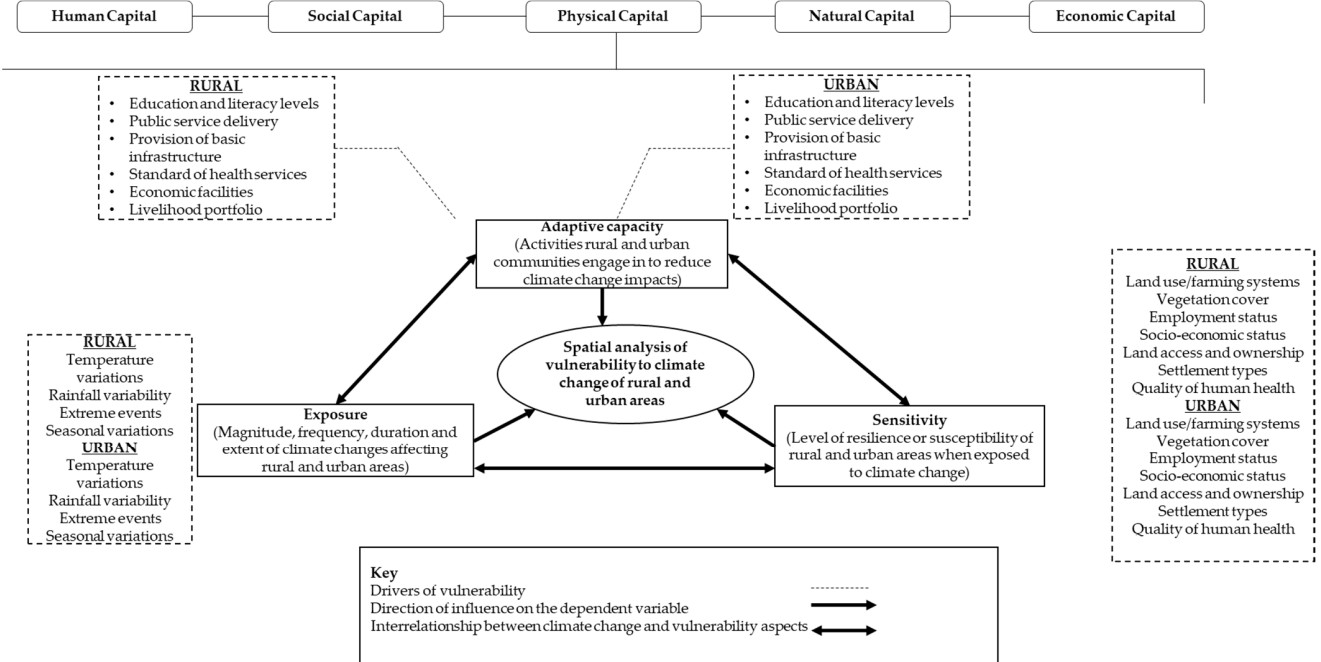

**Figure 3.** Conceptual framework of differences in vulnerability to climate change between rural and urban areas. Source: Authors' conceptualisation.

To holistically understand the differences in climate change vulnerability, the three aspects of vulnerability (exposure, sensitivity, and adaptive capacity) must be captured within the Sustainable Livelihood Approach. The Sustainable Livelihood Approach encompasses five pillars or livelihood assets: human, social, physical, natural, and economic capital. The Sustainable Livelihood Approach or framework was used for this review to assess the cross-sectoral differences in vulnerability to climate change between the rural/urban ecosystems concerning the direct impact on human wellbeing. The presented conceptual framework enables a socio-economic analysis within vulnerability-to-climate-change spatial modelling. Further, the three aspects of vulnerability (exposure, sensitivity, and adaptive capacity) are mapped, taking account of their indicators or drivers within the Sustainable Livelihood Approach.

This review hypothesises that whilst the rural and urban areas are all vulnerable to climate change, there are differences between these two ecosystems. Thus, exposure, sensitivity, and adaptive capacity are driven by various factors within each specific ecosystem based on the associated climate change drivers. For example, we expect the exposure aspect to vary from rural to urban areas based on temperature variation, rainfall variability, frost occurrence, cold spells, frequency of droughts and floods, heatwaves, and hailstorms. One can assume that urban areas are more likely to be exposed to temperature and heat waves due to the increased industrialisation and urban island buildings. On the other hand, rural

areas can also be hypothesised to be more vulnerable to droughts and floods due to less or inadequate infrastructure such as water reservoirs and building infrastructure, respectively. The same differences are expected regarding how resilient the rural and urban ecosystems are, accounting for differences in the influence of land use/farming systems, the impact of vegetation cover on land degradation, employment status of household heads, the socio-economic status of households, the influence of land access and ownership, quality of settlements, influence of seasonal variations, and human health.

Furthermore, whilst the lack of coping activities can be associated with the rural areas, there is also a growing concern for the marginalised or informal settlements in urban areas. Thus, we expect the adaptive capacity to also vary in rural and urban areas primarily driven by differences in factors such as education and literacy levels, service delivery, provision of basic infrastructure, the standard of health services, economic facilities, and livelihood portfolios. Although we expect differences in climate change vulnerability between rural and urban areas, it would not be surprising to find complex relationships and interactions between sectors within each ecosystem that may lead to unexpected patterns of vulnerability. This conceptual framework (Figure 3) underpins/guides our analysis of the reviewed articles within the literature.

## 3. Results and Discussion

### 3.1. Exposure of Rural and Urban Areas to Climate Change Stressors in South Africa

The review identified 14 key climate change stressors that exacerbate exposure in rural and urban areas in South Africa. These were grouped into four main themes: temperature variations, shifts in rainfall patterns, extreme events, and seasonal changes. Table 2 presents a spatial analysis of exposure to climate stressors between rural and urban areas of South Africa as established from the reviewed articles, while Figure 4 shows the frequency of reviewed articles per key climate change stressor. Both the urban and rural areas in South Africa are exposed to extreme events, particularly droughts, floods, heatwaves, cold spells, and hailstorms. A higher proportion of the reviewed articles cited extreme events as the leading climate change stressor exacerbating exposure of rural and urban areas in South Africa, largely in the form of extreme flooding and droughts. However, differences are noted with exposure to drought, mainly in rural areas, while urban areas are more exposed to floods. This finding tallies with Winsemius et al. [24]. They noted that rural areas have the strongest exposure to droughts while urban areas are disproportionately exposed to floods. This finding can be attributed to the nature and characteristics of urban centres, such as rapid urbanisation, inadequate drains, poor waste management, and infrastructure failure, as observed by Aisedu [25].

Temperature variations manifest as changes in temperatures, increased temperature, and extreme heat. We noted that specific indicators of climate stressors for temperature variations vary between rural and urban areas. Rural areas are mainly exposed to increased temperatures, while extreme heat is primarily experienced in urban areas. Another distinction was noted in the locations exposed to temperature variations between rural and urban areas. In rural areas, arid and semi-arid regions are largely exposed to temperature variations. Limpopo, KwaZulu-Natal, and Eastern Cape Provinces, which form large rural areas with agriculture as their main economic activity, are highly exposed to temperature variations. In urban areas, the regions in sub-humid climates are largely exposed to temperature variations. The Western Cape and Gauteng Provinces, which are highly urbanised, are also exposed to temperature variations. Our findings concur with Hu et al. [26]. They sought evidence of the rural–urban disparity in temperature–mortality relationships in China and found that rural residents were more prone to hot temperatures. For urban areas, findings may be attributed to rapid urbanisation prevalent in urban areas, as echoed by Chapman et al. [27]. They indicated that urban areas are more exposed to extreme heat episodes. Exposure increases due to the Urban Heat Island (UHI) intensification associated with urbanisation and high population densities compared to rural areas.

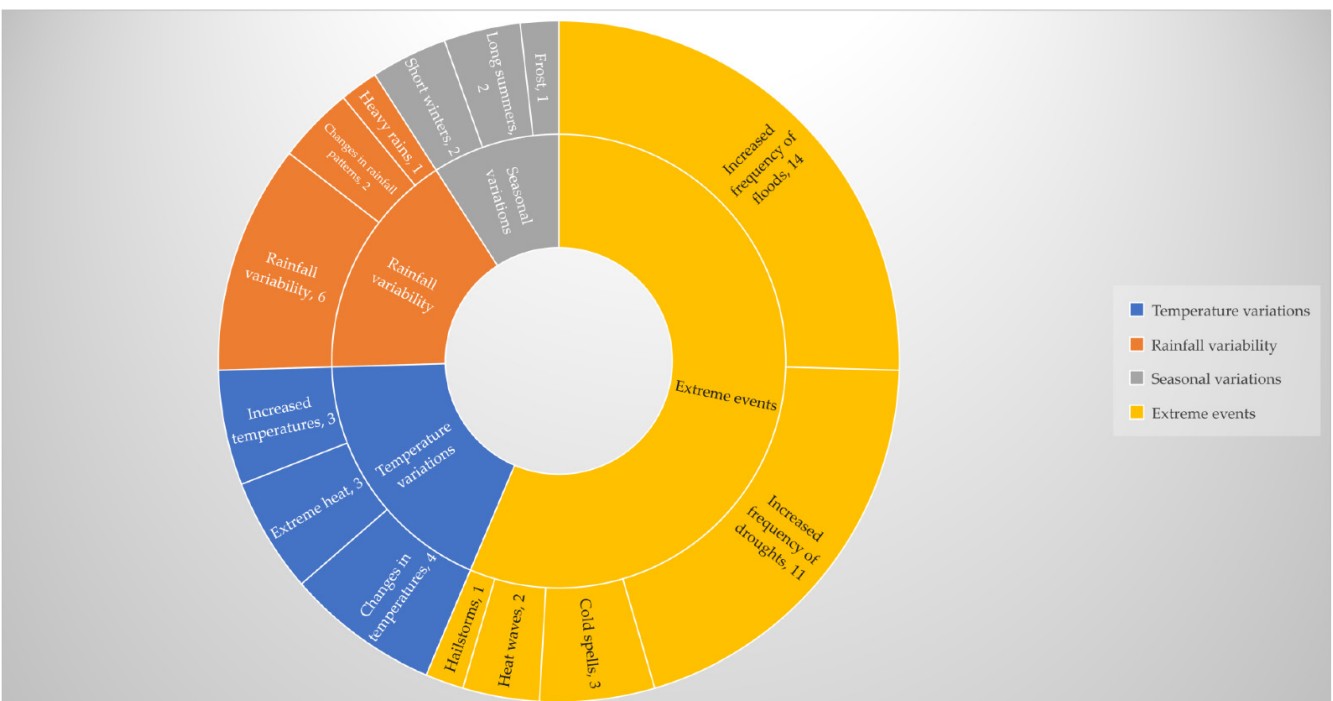

**Figure 4.** Climate change stressors exacerbating exposure of urban and rural communities in South Africa. Source: Literature review analysis for the period between 2010 and 2021 (South Africa).

Regarding rainfall variability, the leading indicators for shifts in rainfall patterns identified in the review are rainfall variability, heavy rains, and changes in rainfall patterns. The review established that exposure to rainfall varies between the unit of analysis considered in different studies. Regarding the unit of analysis, farming households are largely exposed in rural areas. On the other hand, government sectors such as public health, water, and sanitation are highly exposed in urban areas. Our findings for rural areas align with the evidence in several research studies across Africa, especially in sub-Saharan Africa (SSA), on limiting factors for agricultural growth. For example, in Ethiopia, rainfall variability was a limiting factor in the uptake of agricultural credit [28]. This situation implies that farming households will continually be exposed to rainfall variability without the necessary means (capital) to reduce exposure.

Seasonal variations manifest as long summers, short winters, and increased frost occurrences. Long summers and short winters are typical in both rural and urban areas, while high frost occurrences were noted in rural areas. Overall, although there are differences between rural and urban areas of South Africa in terms of exposure to climate change, there is a thin line between the extent of the exposure.

**Table 2.** Climate change stressors exacerbating exposure of urban and rural areas in South Africa.

| Climate Stressors | Descriptors of Climate Stressors | Location and Source | | | Summary of the Findings |
|---|---|---|---|---|---|
| | | **Rural** | **Urban** | **Both Rural and Urban** | |
| Temperature variations | Fluctuating rate of temperature and long-term shifts in temperatures | Hitayezu et al. [29]; Ofoegbu et al. [30]; Goldin et al. [31]; Oosthuizen [32] | Jimoh et al. [33]; Orimoloye et al. [34]; Hlahla & Hill [35] | Gbetibouo et al. [36] Stadler [37] Jarganath et al. [38] | Temperature variations are common in both rural and urban areas. In rural areas, temperature variations are mainly experienced in arid and semi-arid regions. In urban areas, variations are experienced in sub-humid climates. |
| Rainfall variability | High inter-annual variability | Nyahunda et al. [39]; Jimoh et al. [33]; Hosu et al. [40]; | Williams et al. [41] | Gbetibouo et al. [36] Stadler [37] | Both rural and urban areas are exposed to rainfall variability. In rural areas farming households are more exposed. Public health, water, and sanitation services in urban areas are more at risk. |
| Extreme events | High incidence and frequency of extreme events such as droughts and floods | Ncube et al. [42]; Nembilwi et al. [43]; Munyai et al. [44]; Shisanya & Mafongoya [45] Ofoegbu et al. [30]; Udo [46]; Oosthuizen [32]; Goldin et al. [31]; Shackleton et al. [47]; | Orimoloye et al. [34]; Membele et al. [48]; Williams et al. [41]; Hlahla & Hill [35] | Gbetibouo et al. [36] Stadler [37] | Rural and urban areas in South Africa are exposed to droughts and floods. Differences are noted with exposure to drought, mainly in rural areas, while urban areas are more exposed to floods. Hailstorms are more common in urban areas. |
| Seasonal variations | Increased variations in temperature and rainfall between seasons | Hitayezu et al. [29] Goldin et al. [31] | Hlahla & Hill [35] | | Long summers and short winters characterised by cold spells and frost occurrences are common in both rural and urban areas. |

Source: Literature review analysis for the period between 2010 and 2021 (South Africa).

Ge et al. [49] observed that most urban residents still rely on primary industries such as farming and fishing. This finding is the same for urban residents in South Africa. Hence, urban residents are more or less similar to rural residents in terms of exposure to climate change as these activities are primarily affected by climate change stressors.

### 3.2. Sensitivity to Climate Change in Rural and Urban Areas in South Africa

The sensitivity analyses conducted in this review in rural areas largely focused on households, farming systems, ecological zones, communities, and villages. On the other hand, the sensitivity analyses conducted in urban areas were largely on informal settlements owing to a dearth of literature or research on sensitivity to climate change in formal settlements. Some reviewed articles analysed sensitivity for urban and rural areas at the provincial and household levels. Table 3 shows the results, reflections, and insights revealed by the sensitivity analyses conducted in rural and urban areas of South Africa.

Results of the review suggest that households in rural areas are more sensitive to climate change stressors than those in urban areas. This observation is based on findings by Ncube et al. [42], Jimoh et al. [33], and Stadler et al. [37]. Ncube et al. [42] posit that rural households in Limpopo and Eastern Cape are directly and indirectly sensitive to climate change, respectively. Jimoh et al. discovered that households in selected towns semi-arid towns, including Tzaneen, Nkowankowa, and Hoedspruit, were not sensitive to climate change impacts. Stadler et al. [37] shared the same view. They showed that in the Eastern Cape, households located in formal residential urban areas of Lessyton were less sensitive than households located in rural areas of Gatyana in the same province.

Stadler et al. [37] explained that the differences in sensitivity between rural and urban areas could be attributed to several factors, such as differences in household ownership, with households in rural areas governed by traditional structures. In contrast, those in urban areas were individually governed. Such arrangements have a bearing on how quickly a household can respond to climate stressors. If a household is individually governed, it is easy to decide and act immediately as opposed to households traditionally governed with structures and protocols to consider before making a decision. Stadler et al. [37] further explained that the differences between the layout of households in rural and urban areas are another reason for the differences in sensitivity. Unlike in urban areas, households in rural areas are scattered across the landscape with poor roads isolated from major transport routes. The layout in rural areas makes it difficult for disaster risk management teams to reach some areas. Hence, people in such isolated parts are more sensitive than those in urban areas where there are well-established transport route systems. Apart from that, households in rural areas are mainly traditional homes, rondavels with thatched roofs and no electricity. Some of the materials used to construct these homes are of low quality (mud as opposed to bricks). Hence, they are more sensitive to climate change impacts than households in urban areas built with strong and good quality material.

**Table 3.** Major results, reflections, and insights on sensitivity to climate change in rural and urban areas of South Africa.

| Rural | | Urban | | Both Rural and Urban | |
|---|---|---|---|---|---|
| **Source** | **Major Results** | **Source** | **Major Results** | **Source** | **Major Results** |
| Ncube et al. [42] | Households in Alice, Eastern Cape are more sensitive than in households in Lambani, Limpopo. | Jimoh et al. [33] | Most households across selected towns in Mopani District were not sensitive to climate change stressors. | Gbetibouo et al. [36] | The most sensitive provinces are Limpopo, Kwazulu-Natal, and Eastern Cape. |
| Hitayezu et al. [29] | Farming systems with small-scale farming, low irrigation rates, and that are prone to land degradation are highly sensitive. Diversified crop systems have high resilience. | Williams et al. [41] | The sensitivity of informal settlements to flooding is influenced by levels of education, access to public services, provision of basic infrastructure, and health standards. | Stadler [37] | Formal residential areas (suburbs) with high household ownership levels, green open spaces, vegetation, and commercial or mixed land uses are less sensitive. |
| Ofoegbu et al. [30] | Forest-based communities have uneven sensitivity due to uneven exposure to various types and magnitudes of stressors. | Orimoloye et al. [34] | Human health is extremely sensitive to extreme weather. | Stadler et al. [37] | People living in rural areas of Gatyana are more sensitive to HIV/AIDS and climate change than people in urban areas owing to the diseases' associations with marginalised communities. |
| Goldin et al. [31] | Women and girls are more sensitive than males. | Hlahla and Hill [35] | Socio-economically marginalised urban communities are highly sensitive to seasonal variations, drought, heat waves, cold spells, hailstorms, and floods. | Chersich et al. [49] | The most sensitive populations in South Africa are women, fishing communities, subsistence farmers, and those living in informal settlements. |
| Shackleton et al. [47] | Households' sensitivity is a function of livelihood activities, poverty levels, and asset holdings. | Wedepohl [50] | Sensitivity to climate change stressors differs between informal and formal settlements. | Chikulo [51] | Differences in sensitivity are noted between women in electrified urban homes and rural women with non-electrified homes. |
| Udo [46] | Women's sensitivity to floods is increased by poverty, inequality, marginalisation, lack of access to loans and insurance, poor quality of houses and other infrastructure, and lack of knowledge and education. | - | - | - | - |

**Table 3.** *Cont.*

| Rural | | Urban | | Both Rural and Urban | |
|---|---|---|---|---|---|
| **Source** | **Major Results** | **Source** | **Major Results** | **Source** | **Major Results** |
| Oosthuizen [32] | Farming systems are sensitive to climate-induced financial vulnerability. | - | - | - | - |
| Shackleton and Cobban [52] | Rural women are highly sensitive to climate change due to reliance on ecosystem services, low income, labour constraints, and poor health. | - | - | - | - |
| Abayomi [53] | People living with HIV/AIDS are more sensitive to climate change stressors and are at a health disadvantage in a changing climate. | - | - | - | - |
| Mugambiwa and Tirivangasi [54] | Poor rural communities in South Africa are immensely susceptible to climate change owing to a lack of livelihood assets leading to increased hunger and malnutrition. | - | - | - | - |
| Munyai et al. [44] | The nature of soil and type of dwelling are the most important factors influencing sensitivity to climate change in rural areas. | - | - | - | - |
| Own critical analysis | Rural: Households, farming systems, ecological zones, communities, and villages in rural areas are sensitive to climate change. The type and nature of farming systems significantly bear on sensitivity levels. "Forest-based" rural communities are differentially sensitive due to different forest types. | | Urban: Sensitivity to climate change varies between formal and informal settlements in urban areas. Informal settlements are more sensitive to climate change. "Urban poor" communities are highly sensitive because most are socio-economically marginalised. | | Both: Female-headed households are more sensitive to climate change than male-headed households. |
| A spatial perspective | Households in rural areas are more susceptible to climate change than urban areas. Sensitivity at the household level varies between rural and urban areas. Rural communities are unevenly sensitive at the community level, while urban communities are highly sensitive. | | | | |

Source: Literature review analysis for the period between 2010 and 2021 (South Africa). (-) denotes No data.

Despite these findings, sensitivity at the household level varies within rural and urban areas. A study conducted by Ncube et al. [42] found that households in the rural area of Alice, Eastern Cape, were directly exposed to sensitivity. At the same time, those in Lambani, Limpopo were indirectly susceptible. This discovery is due to several factors. As established from the review, the type and nature of livelihood activities practised by households in rural areas result in variations in sensitivity and resilience to climate change. For example, Shackleton et al. [47] established that households that derive most of their income from government grants and self-employment were more susceptible than those from formal employment. This explanation is in line with Mildrexler et al. [55]. They posited that low-income households, including the less educated and unemployed, are more sensitive to climate change stressors than high-income households with high-income jobs. Similarly, Shackleton and Cobban [52] illustrate that households' reliance on ecosystem-based livelihoods increases their sensitivity to climate change more than those that rely on economic livelihoods. Variations in levels of poverty that exacerbate associated variables such as marginalisation and inequality, among others, increase household sensitivity to climate change. This view was also confirmed by Udo [46], who established that differences in poverty levels among different households are a factor that results in different sensitivity levels to climate change.

Variations within farming systems located in rural areas were also noted. Hitayezu et al. [29] established that sensitivity analyses of farming systems located in rural areas of KwaZulu-Natal showed that the type and nature of farming systems significantly affect sensitivity levels. Farming systems with high rates of small-scale farmers, low irrigation rates, and areas prone to land degradation are highly susceptible to climate change. Further, diversified farming systems have lower sensitivity to climate change stressors than non-diversified systems. Oosthuizen [32] showed that the sensitivity of farming systems to climate-induced financial vulnerability varies between farming systems. In the Western Cape province, the sensitivity of farming systems to climate-induced financial vulnerability varied, with farming systems in Vrendedal found to be highly susceptible. In contrast, those in Moorreesburg were found to be marginally sensitive.

In urban areas, variations in sensitivity have also been identified between formal and informal settlements. A study conducted by Wedepohl [50] in Durban, KwaZulu-Natal, found that sensitivity to climate change varies between formal and informal areas due to variations in the five main types of capital: human, social, biophysical, economic, and institutional. Significant differences in sensitivity between the Westville formal residential area and the Quary Road West informal settlement were due to the human and economic capital being comparatively higher in the former.

Rural communities are unevenly sensitive at the community level, while urban communities are highly sensitive. However, it is imperative to note that the communities mentioned in the studies considered for review are different. Sensitivity analyses in rural areas were for "forest-based" communities, while the focus was on "urban poor" communities in urban areas. Despite this fact, the emphasis was on the unit of analysis for the comparison in this review. It was established that differential susceptibility of "forest-based" rural communities resulted from forest types that vary between communities. Some forest types may be more sensitive to climate change than others. On the other hand, the high sensitivity of "urban poor" communities was because most are socio-economically marginalised.

A gender-based lens on sensitivity analyses shows that women in rural and urban areas are highly susceptible to climate change stressors. Female-headed households are generally more susceptible to climate change shocks than male-headed households. This finding corresponds to the traditional views of the binary male–female view of the gender dimension of susceptibility to climate change that women are passive victims of climate change [56]. However, Shackleton et al. [47] posit that although women may appear more susceptible to climate change than men, men are also susceptible because they rely more on livestock production as their main source of income, which is sensitive to climate change impacts. However, despite this being the case, Chikulo [51] explored the gender, climate

change, and energy linkages in South Africa and observed that there are differences in sensitivity to climate change among women. The author suggests that rural women and those in poor urban areas without electricity are sensitive to climate change. The author also distinguished between urban women with electrified homes and those without and stated that the former were less sensitive to climate change. This observation could be because women with non-electrified homes are more likely to bear the brunt of high temperatures and heat episodes as they travel long distances in search of firewood than those with electrified homes. In this regard, Babugura [57] offers insights on the role of institutions in reducing gender-related sensitivity to climate change and illustrates that institutions can empower both men and women in reducing sensitivity to climate change impacts.

Abayomi [53] and Stadler [37] provided a health-related view on sensitivity to climate change. Abayomi [53] illustrated that rural people living with HIV/AIDS are more sensitive to climate change impacts putting them at more risk in a changing climate. Similarly, Stadler [37] demonstrated that HIV/AIDS is more prevalent in rural sites of Gatyana than in urban areas of Lessyton, owing to the disease's associations with marginalized communities. This finding could be explained by the fact that people in urban areas have more access to health facilities than those in rural areas, where health facilities may be located far away and in isolated places. Mugambiwa and Tirivangasi [54] also gave insights into the impact of susceptibility to climate change on food security and nutrition issues. They showed that poor communities in South Africa are highly susceptible to climate change posing a major risk to food security and nutrition. This has resulted in increased hunger and malnutrition.

### 3.3. Adaptive Capacity of Rural and Urban Areas in South Africa

Findings of the review show that adaptive capacity is generally lower in rural areas than in urban areas. This evidence is based on a study by Gbetibouo et al. [36]. They observed high adaptive capacity in the Western Cape Province, which is considered urban with well-developed infrastructure, high levels of literacy and income, and low levels of unemployment and HIV prevalence. In contrast, low levels of adaptive capacity were observed in the more rural Limpopo Province, with a high share of small-scale farmers that rely on rain-fed agriculture, high population density, high unemployment, low literacy levels, and poor infrastructure. The findings tally with observations made by Ribot [58] that adaptive capacity varies systematically along prevailing fault lines of inequality and social exclusion. It is imperative to recognise that rural and urban residents' socio-economic status in South Africa is not on par. For example, unlike in urban areas, wealth is often expressed in the form of land and cattle ownership in rural areas [59]. These assets are not easily converted into cash required to enhance adaptive capacity. Hence, disparities in coping with climate change stressors vary between rural and urban residents. Table 4 shows the results of an analysis of the adaptive capacity of rural and urban areas in South Africa.

The review found that adaptive capacity is influenced by the five main types of capital: human, physical, financial, natural, and social capital in both rural and urban areas. This observation was made in several articles reviewed, including Ncube et al. [42], Oosthuizen [32], and Wedepohl [50]. This finding concurs with Heltberg and Bonch-Osmolovkiy [60]. They noted that the five main types of capital are the drivers of adaptive capacity. As such, households with less capital have a lower adaptive capacity than those with high capital.

**Table 4.** Major results on the adaptive capacity of rural and urban areas in South Africa.

| Rural | | Urban | | Both | |
|---|---|---|---|---|---|
| Source | Major Results | Source | Major Results | Source | Major Results |
| Ncube et al. [42] | The adaptive capacity of households is influenced by human, physical, financial, natural, and social capital. | Jimoh et al. [33] | A significant proportion of sampled households (76.2%) could adapt to climate change impacts. | Gbetibouo et al. [36] | The Western Cape province has the highest adaptive capacity due to its well-developed infrastructure, high levels of literacy and income, low levels of unemployment, and HIV prevalence. |
| Hitayezu et al. [29] | Adaptive capacity is limited by inadequate access to infrastructure, rural exodus, skills shortages, poor health status, and lack of cooperation among farmers. | Williams et al. [41] | Governance shapes education standards, delivery of public services, provision of basic infrastructure, and the standard of health and economic facilities, which influence adaptive capacity in informal settlements. | - | - |
| Ofoegbu et al. [30] | The adoption of adaptation measures in rural communities is appreciated; however, capacity is often insufficient to maintain resilience and sustainability. | Hlahla and Hill [35] | The majority do not have the means to respond to climate change impacts due to a lack of education and the belief that nothing can be done to deal with climate change. | - | - |
| Goldin et al. [31] | Political freedom, economic facilities, social opportunities, and protective security are necessary for women to enhance their adaptive capacity. | Membele et al. [48] | People in informal settlements have differentiated adaptive capacities. Indigenous knowledge strengthens adaptive capacities in informal settlements. | - | - |
| Shackleton et al. [47] | Gender-based violence is cited as one factor diminishing women's adaptive capacity in rural areas. | Wedepohl [50] | The interrelatedness of the available types of capital impacts the resilience and adaptive capacity of informal and formal settlements. | - | - |
| Udo [46] | Although women demonstrated "agency" in adapting to floods, their adaptive capacity is often limited by poverty, increased levels of abuse, and lack of political connections. | Roberts and O'Donoghue [61] | The adaptive capacity of the city of Durban is low due to several factors, including high rates of poverty and unemployment, and a lack of skilled human resources to carry out adaptation planning and implementation, among other challenges. | - | - |

**Table 4.** *Cont.*

| Rural | | Urban | | Both | |
|---|---|---|---|---|---|
| Source | Major Results | Source | Major Results | Source | Major Results |
| Oosthuizen [32] | Biophysical factors are important in improving the adaptive capacity of farming systems. | Ziervogel et al. [62] | Major factors reducing adaptive capacity at the Municipal level are inadequate communication between scientists, policymakers, and practitioners, a lack of coordination between different scales of operation, and a lack of human capacity to implement policy. | - | - |
| Quinn et al. [63] | Social, economic, and political conditions shape adaptive capacity; hence, adaptation processes should not be viewed in isolation, but a holistic approach should be adopted to account for all those factors. | - | - | - | - |
| Nyahunda et al. [39] | Gender inequalities manifesting through unequal land and property rights, discrimination in the decision-making process, and unequal sharing of burdens undermine women's adaptive capacity. | - | - | - | - |
| Own critical analysis | Rural: Adaptive capacity is influenced by the five main types of capital: human, physical, financial, natural, and social capital in both rural and urban areas. Rural communities make use of adaptive measures, but their capacity is often insufficient to match the ever-increasing climate changes. | | Urban: Urban areas of South Africa have shown differentiated adaptive capacity. | | Both: Highly developed regions have a higher adaptive capacity than less developed regions. |
| A spatial perspective | Although climate change is not gender-neutral, women are assumed to have less adaptive capacity. Poverty is the greatest limitation in adapting to climate change for both females and males. Women do not always have a say in adaptation decisions. This situation makes them more dependent on men's decisions and more vulnerable to climate change impacts. | | | | |

Source: Literature review analysis for the period between 2010 and 2021 (South Africa). (-) denotes No data.

The major limiting factors of adaptive capacity in rural areas of South Africa, as noted by Hitayezu et al. [29], include poor infrastructure, skills shortages, and a lack of cooperation among farmers. Infrastructure has been identified as one of the generic determinants of adaptive capacity in the existing literature [64]. Infrastructure such as roads and irrigation equipment aid adaptation in farming communities [65]. However, these have deteriorated in many rural parts of South Africa due to a lack of or no maintenance, unlike in urban areas where roads are well established and constantly maintained. Literacy levels, knowledge of climate information, farming experience [65], and managerial ability [66], considered a proxy for skills, also influence adaptive capacity. Nonetheless, the farming communities in rural areas of South Africa are found wanting in all these factors and hence have a low adaptive capacity.

The review findings noted and appreciated the adoption and use of adaptation measures in rural areas of South Africa. However, it was established that capacity is often insufficient to match the ever-increasing climate changes in South Africa and maintain sustainability and resilience [30]. This trend is noted in other developing countries as well. For example, Huq et al. [67] also noted declining adaptation capacities among agricultural communities in rural parts of Bangladesh with fewer livelihood resources, resulting in longer recovery processes. The main factor that led to a decline in adaptive capacities in Bangladesh was the strong magnitude of climatic events, just like in South Africa.

Urban areas of South Africa have shown differentiated adaptive capacity contrary to the rural areas. This observation was established in a study by Membele et al. [48] in informal settlements countrywide in urban areas. Unlike in rural areas of South Africa, Williams et al. [41] noted that factors such as governance, delivery of public services, and provision of basic infrastructure increase the adaptive capacity of urban residents. Although Williams et al. [41] referred to the adaptive capacity of residents in informal areas, these factors are also critical for other urban residents elsewhere, as shown by Wedepohl [53] who emphasised the interrelatedness of different types of capital enhancing adaptive capacity.

Roberts and O'Donoghue [61] observed that the socio-economic characteristics of the population in metropolitan cities have a bearing on adaptive capacity. Their case study for the city of Durban showed that it is plagued by high levels of poverty and unemployment, leading to its low adaptive capacity. Ziervogel et al. [62] offer insights on adaptive capacity in urban areas at the municipal level, seeking to identify factors impeding and facilitating adaptation in the water supply management sector for Cape Town. Their analysis offered the perspective that, to scale up adaptation in cities, there is a need to strike a balance in addressing factors impeding and those facilitating it.

The review also established gender dynamics by Goldin et al. [31], Shackleton et al. [47], Udo [46], and Williams et al. [41]. Although climate change is not gender-neutral [68], women are considered a highly vulnerable group and assumed to have a lesser adaptive capacity [69] than men. Udo [46] applauds women's demonstrated "agency" in adapting to floods. However, the same author noted that adaptive capacity among women is often limited by poverty, increased levels of abuse, and a lack of political connections.

Poverty has been listed as the greatest limitation in adapting to climate change, not only for women but for men as well. For example, Heltberg and Bonch-Osmolovkiy [60] reported that the poor (men and women) are least equipped to adapt to climate change impacts. Serumaga-Zake and Naude [70] found that education and household size were the main determinants of household poverty in the Northwest Province. This discovery is aligned with the finding that rural areas with low literacy levels and bigger household sizes have lower adaptive capacity. Gender inequalities exacerbate increased levels of abuse and gender-related violence. Serumaga-Zake and Naude [70] observed a lower degree of gender discrimination in urban areas than in rural areas. This situation is possibly because women in urban areas have more access to platforms that advocate for women's empowerment than their rural counterparts [71]. Phan et al. [26] share another view that rural women choose to remain silent to avoid tension in the family. This stance contradicts their urban counterparts, who are more empowered to speak out and fight for their rights. This finding may imply

that rural women do not always have a say in adaptation decisions. This situation makes women more dependent on men's decisions and more vulnerable to climate change impacts. Apart from that, Phan et al. [26] observed that men dominate political organisations. This situation allows men to have power and control over resources at the community level, increasing their adaptive capacity relative to their female counterparts. Women generally lack the political power and freedom to access economic facilities and social opportunities that may enhance their adaptive capacity, as observed by Goldin et al. [31].

## 4. Conclusions

This review aimed to establish how vulnerability to climate change varies between rural and urban areas in South Africa and to understand the sources of these variations. The review was guided by the IPCC conceptualisation of vulnerability regarding exposure, sensitivity, and adaptive capacity. We drew insights from the Hazards of Place (HOP) model. The idea of "place" was borrowed to provide a spatial perspective in understanding the dynamics of vulnerability between rural and urban areas in South Africa.

As expected or hypothesised, we found differences in vulnerability to climate change between rural and urban areas. Differences were noted based on households' exposure, sensitivity, and adaptive capacity across farming systems, settlements, provinces, and agro-ecological regions. The review shows that rural areas in South Africa are more vulnerable than urban areas in all three vulnerability aspects: exposure, sensitivity, and adaptive capacity. This revelation may be attributed to many factors that distinguish rural from urban areas. These include but are not limited to differences in settlement types, population densities, and common livelihood and income-generating activities that vary between rural and urban areas. Rural communities rely more heavily on climate-sensitive resources, for example, agriculture (both crop and livestock farming) and natural resources (land, wildlife, forestry, and water sources, among others) than urban communities. In rural areas, variations are pointed out between provinces, farming systems, agro-ecological zones, villages, and households. In urban areas, variations are noted between settlement types, communities, and households.

The review also established that differences in vulnerability between rural and urban areas emanate from differences stemming from people's socioeconomic status, demographic traits, social networks, access to resources, basic infrastructure, and political power. The review concludes that vulnerability varies with location and requires place-based analyses to develop relevant policies that enhance resilience and adaptation.

Findings of the review suggest that differential vulnerabilities to climate change are a result of multi-dimensional disparities and unequal development pathways between urban and rural areas, leading to varying degrees of climate change risks. Understanding the unique cultural, socio-political, and economic scenarios of different urban and rural communities is therefore paramount. This would enable the identification of specific vulnerabilities for rural and urban communities and enhance governments and other stakeholders to counter them, making communities resilient over time. Thus, applying and implementing just transition principles through collective and participatory decision-making processes is an effective way of integrating equity principles into policies to address differential urban and rural communities' vulnerabilities.

Based on the review's findings, enabling policies are required for both urban and rural areas to advance climate action that decreases vulnerability. Policy initiatives should consider a breath of focus to lessen vulnerability in rural and urban areas. Initiatives should be streamlined in all three facets of vulnerability—exposure, sensitivity, and adaptive capacity. As a result of the interaction between these three factors, policymakers should strive to develop cross-sectional and transboundary solutions that minimize vulnerability to climate change while also increasing adaptive capacity in rural and urban areas. Early warning systems and climate expert services are crucial for rural areas that significantly rely on climate-sensitive livelihoods. For urban areas, policies should pursue climate action that generates green economic growth, employment creation, and increases the well-being of the

people to increase adaptive capacity. Overall, supportive national policies are required to ensure that rural and urban initiatives have adequate financial resources for adaptation and mitigation actions to reduce vulnerability. The South African government should ensure that the National Climate Change Adaptation Strategy is well-aligned with rural and urban communities' context-based vulnerabilities.

As a recommendation for further studies, the review noted an imbalance in research on sensitivity to climate change between formal and informal settlements. Therefore, further research is required to explore the differences in vulnerability between the formal and informal settlements in South Africa. Considering the new global and current trends in emigration to towns and cities in South Africa, we also noted a gap in the literature on vulnerability to climate change impacts between urban and rural areas. In this regard, further studies are required to explore the spatial effects on vulnerability.

A limitation of this review is the narrow focus on South Africa with a relatively small sample size due to the scarcity of literature. Therefore, we conclude with caution that although we observed some differences in vulnerability to climate change between rural and urban areas, it is recommended that further comparative analysis of broadened scope be undertaken. The recommendation is that such a comparative analysis could span across regions in Southern Africa or Africa with different levels of economic development to provide more robust conclusions.

**Author Contributions:** Conceptualization, L.Z., D.S.K., M.S. and K.N. Original draft preparation, D.S.K. Literature search and screening process, D.S.K. Inputs, comments, scientific validity and rewriting the final submitted manuscript, L.Z., M.S. and K.N. All authors have read and agreed to the published version of the manuscript.

**Funding:** This review paper is part of a post-doctoral fellowship of the second author under the supervision of the first author. The post-doctoral fellowship is funded by the SAF-ADAPT project headed by the African Climate Development Initiative (ACDI) at the University of Cape Town, partnering with the Risk and Vulnerability Science Centre at the University of Fort Hare. The authors are grateful for the support.

**Acknowledgments:** The authors are grateful to the staff and colleagues at Risk and Vulnerability Science Centre, University of Fort Hare, for their support.

**Conflicts of Interest:** The authors declare no conflict of interest.

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
