# Peer review of "An Analysis of the Differences in Vulnerability to Climate Change: A Review of Rural and Urban Areas in South Africa"

_climate, doi:10.3390/cli10080118_

Round 1

Reviewer 1 Report

The revised manuscript is improved. 

I would suggest the authors mention " differences between rural and urban areas" rather than "spatial variability" throughout the manuscript. The results/discussion are mostly true regardless of the spatial location/proximity of rural and urban areas. 

Author Response

Point 1: I would suggest the authors mention " differences between rural and urban areas" rather than "spatial variability" throughout the manuscript. The results/discussion are mostly true regardless of the spatial location/proximity of rural and urban areas.

Response 1: Thank you very much for the suggestion. As suggested, the phrase “spatial variability” was replaced with “differences between rural and urban areas” throughout the manuscript.

Reviewer 2 Report

Dear Authors,

 Thank you for your exciting work. The authors provide a spatial analysis of vulnerability to climate change in South Africa. The paper focuses on the vulnerability to climate in urban and rural areas. The paper is well-written – I enjoy reading it. However, as I noted in the previous review, “the major concern that I have is that the paper relies on some 8 and 6 papers that analyzed the vulnerability to climate for urban and rural, respectively. The small sample size likely leads to biased results. Thus, this deficiency makes the main contribution of this paper is questionable. The major finding of this paper – there are differences in vulnerability to climate change between rural and urban areas – is predictable.” As the paper stands now, it still relied on the 8 and 6 papers (see table 1).   

I strongly recommend the authors to wider the search to find more papers so the sample size could be bigger. Alternatively, the paper can be re-oriented to also consider the vulnerability across regions/geographical locations in Southern Africa (not just South Africa county) -  an additional section or two sections analyzing the vulnerability across regions/geographical locations could be sufficient. The comparisons between South African .vs counties that have different levels of economic development are worth noting.   

Author Response

Point 1: However, as I noted in the previous review, “the major concern that I have is that the paper relies on some 8 and 6 papers that analyzed the vulnerability to climate for urban and rural, respectively. The small sample size likely leads to biased results. Thus, this deficiency makes the main contribution of this paper is questionable. The major finding of this paper – there are differences in vulnerability to climate change between rural and urban areas – is predictable.” As the paper stands now, it still relied on the 8 and 6 papers (see table 1).  

Response 1: Thank you very much for the comments. However, while the authors acknowledge the limitation in the sample size, the authors wish to express that the review was not based on “some 8 and 6 papers” as stated by the reviewer, but 30 reviewed articles. The sample size was informed by the search and screening criteria (see Figure 1). The review aimed to explore spatial differences (which has since changed to “differences in vulnerability to climate change” as recommended by reviewer 1) with a focus on the rural and urban areas in South Africa. Therefore, only literature sources and/research conducted in South Africa were sought for inclusion in the review process. Apart from that, the screening process aimed to retain current research to provide an analysis that would portray a recent analysis. Thus, only articles and/or research conducted between 2010 and 2021 were included in the review. However, as suggested by the reviewer in the previous round, the authors conducted the same process again, refined some of the search terms and expanded/ widened the search and found eleven more articles which were included in the analysis. As a result, 30 instead of the initial 19 articles were retained for the analysis (See Table 1 for the breakdown of the literature sources included for the analysis). There are 16 articles for rural, 8 for urban areas and 6 for both rural and urban areas. This brings to a total of 30 articles finally included in the review.

Point 2: I strongly recommend the authors to wider the search to find more papers so the sample size could be bigger. Alternatively, the paper can be re-oriented to also consider the vulnerability across regions/geographical locations in Southern Africa (not just South Africa county) -  an additional section or two sections analyzing the vulnerability across regions/geographical locations could be sufficient. The comparisons between South African .vs counties that have different levels of economic development are worth noting. 

Response 2: Thank you very much for the suggestion. While this is a valid comment, to map the differences between regions and countries with different economic development, the authors have chosen to maintain the review in the context of South Africa, exploring the differences in vulnerability to climate change with a focus on the rural and urban and rural areas. The authors feel that if we reorient focus of the review, it could change the focus altogether. However, we acknowledge this limitation and have highlighted it in the conclusion section. We do also caution on the conclusion that were based on a smaller sample size and recommend the inclusion of a comparative analysis with a broadened scope for the Southern Africa region and perhaps the whole of Africa as a future research Agenda to arrive at more robust conclusions.

Reviewer 3 Report

This is an excellent paper, and I congratulate the authors to bring out this excellent piece.  I would strongly recommend to publish the paper.  Just one minor comment is that to have some policy implications of the findings will enrich the paper. I suggest to include this in the discussion.  Rest is quite fine. 

Round 2

Reviewer 2 Report

I have no further comments to add.

Author Response

This manuscript is a resubmission of an earlier submission. The following is a list of the peer review reports and author responses from that submission.

Round 1

Reviewer 1 Report

This paper aims to explore the main factors regarding vulnerability to climate change in South Africa focusing on differences between rural and urban settlements claiming that we can find there important differences on exposure, sensitivity and adaptive capacity. It is an interesting starting point as wells as an important research issue and the outcomes of this paper are, at least theoretically, of major interest for a multitude of different areas in the world. The paper is well supported by data and scientific literature and written in good standard academic style.The weak point of the paper is the explanation of the methodological analysis. Authors should incorporate more theoretical inputs about the meta-analysis methodology and a bettered detailed explanation of all steps undertaken in that process.

The first paragraph is a little confusing when introducing the three perspectives of vulnerability. It would be much useful to discuss each one a little more in detail (devoting perhaps a line, or at least three or four words) to allow the reader understand better the differences between them. In this version, concepts are intertwined making the argument unclear.

Although I feel the paragraph devoted to the conceptualisation of rural and urban divide as convincing, It would be of use if authors could provide local studies employing the same approach.

The first paragraph of the “Materials and methods” section is confusing, especially regarding the “three-step review” process. The meta-analysis strategy conducted by the authors is well known but it should be in any case better explained and specific bibliography should be used to do so (for example, Grant & Booth, 2009, or PRISMA-P recommendations in Moher et al, 2015). Also the strings used to search for articles should be better clarified: were asterisks used? Only inverted quotations? Were the strings included in the paragraph used exactly like this? Were articles’ titles and abstracts of all sourced articles screened by two independent researchers, as usually the method requires? Was employed any kind of coding (see Gaur & Kumar, 2018)?

Regarding Table 1. I think “class” is probably not the best word to be used there. “Class” in social science usually is linked to “social class”. Maybe “habitat” is a better choice. I am also not convinced about “geographical location” given that all locations are actually located in a geographical space. Maybe “municipalities”?

I am not sure that a definition of theoretical framework is needed.

The “Sustainable Livelihood Approach” should be better explained and introduced (some authors also to back it up). In this version, is, so to speak, coming a bit out of the blue. In the same vein, when the five types of capital are introduced, they should be explained in better detail. Which is the difference between “human” and “social” capital? Which are the differences between “physical” and “natural” capital? Readers should be aware of these diferences before Figure 2.

Figure 2 and 3 should be better edited, maybe diminishing the size of the letters, in order to ease interpretation and reading. Parts of the text are outside the range of the page.

I found Table 2 very useful but I am not sure that “own critical spatial analysis” is actually the best descriptor for the last column because I assume that authors try to summarise there the main conclusions of the articles regarding that kind of climate stressor. If that is the case, probably “own critical analysis” is not fully correct and could be misleading.

On 3.2 section, authors point out that “On the other hand, sensitivity analyses conducted in urban areas were largely on informal settlements”. It seems a limitation of those articles but it is not clearly stated like that.

Maybe a map of South Africa depicting the regions could be of use, especially after Table 3 (which again, I feel very enlighten).

Grant, M. J. & Booth, A. (2009) A typology of reviews: An analysis of 14 review types and associated methodologies. Health Information & Libraries Journal 26(2), 91-108

Gaur, A., & Kumar, M. (2018). A systematic approach to conducting review studies: An assessment of content analysis in 25 years of IB research. Journal of World Business, 53(2), 280-289.

Moher, D., Shamseer, L., Clarke, M., Ghersi, D., Liberati, A., Petticrew, M., ... & Stewart, L. A. (2015). Preferred reporting items for systematic review and meta-analysis protocols (PRISMA-P) 2015 statement. Systematic reviews, 4(1), 1.

Reviewer 2 Report

Dear Authors,

Thank you for your exciting work. The authors provide a spatial analysis of vulnerability to climate change in South Africa. The paper focuses on the differences between vulnerability to climate in urban and rural areas. The paper is well-written – I enjoy reading it. However, the major concern that I have is that the paper relies on some 8 ad 6 papers that analyzed the vulnerability to climate for urban and rural, respectively. The small sample size likely leads to biased results. Thus, this deficiency makes the main contribution of this paper is questionable. The major finding of this paper – there are differences in vulnerability to climate change between rural and urban areas – is predictable.

I strongly recommend the authors to wider the search to find more papers so the sample size could be bigger. Alternatively, the paper can be re-oriented to also consider the vulnerability across regions/geographical locations in South Africa -  an additional section or two sections analyzing the vulnerability across regions/geographical locations could be sufficient.   

Minors

  • Figure 3: the legend got cut off and the texts in the figure are not readable.
  •  Text formatting is not consistency  

Reviewer 3 Report

Thank you for this tightly focused and clear review. I am recommending  it for publication. My only concern, and one which the authors may choose to address, is that with such clear findings I would some indications of the directions for climate adaptation would be forthcoming. In other words, given that literature makes the nature of this problem clear what are potential and prospective solutions? Who needs to do what?

Reviewer 4 Report

#1 Abstract:

“We conclude that vulnerability varies with location and requires place-based analyses to come up with appropriate policies that enhance adaptation.”  Then what is the broader implication of this study?  I suggest authors generalize the results for any administrative boundary (for example, country South Africa)  

#2 Page 2: “Definitions for rural and urban are complex and they vary in South Africa.” This should not be difficult. There should be some government norms/indices that separate urban and rural.

#3

V = f (E, S, A, C)

How do the correlations between E/S/A/C affect the analysis and conclusions? Discuss

Round 2

Reviewer 1 Report

Thanks for your answers

Reviewer 2 Report

Dear Authors,

Thank you for the revision. However, the revision does not seem to address my major concerns. 

The paper needs extensive and throughout revision before sending back to the Journal for re-consideration, perhaps, as a new submission: formating and font consistency; tables got cut-off. I could not read the tables. Again, the paper is in bad shape right now. 

It seems that the paper still draws conclusions based on a relatively small sample size, which might lead to biased results.  

Reviewer 4 Report

I suggest authors address all review comments and make necessary changes in the manuscript. The authors only replied to the review comments but didn't make any effort to change/modify the manuscript accordingly